# Tissue Sheet Engineered Using Human Umbilical Cord-Derived Mesenchymal Stem Cells Improves Diabetic Wound Healing

**DOI:** 10.3390/ijms232012697

**Published:** 2022-10-21

**Authors:** Jingbo Zhang, Xiang Qu, Junjun Li, Akima Harada, Ying Hua, Noriko Yoshida, Masako Ishida, Yoshiki Sawa, Li Liu, Shigeru Miyagawa

**Affiliations:** Department of Cardiovascular Surgery, Osaka University Graduate School of Medicine, 2-2 Yamada-oka, Suita 565-0871, Osaka, Japan

**Keywords:** diabetic wound, human umbilical cord-derived mesenchymal stem cells, PLGA scaffold, inflammation, collagen deposition, wound healing

## Abstract

Diabetic foot ulceration is a common chronic diabetic complication. Human umbilical cord-derived mesenchymal stem cells (hUC-MSCs) have been widely used in regenerative medicine owing to their multipotency and easy availability. We developed poly(lactic-co-glycolic acid) (PLGA)-based scaffold to create hUC-MSC tissue sheets. In vitro immunostaining showed that hUC-MSC tissue sheets formed thick and solid tissue sheets with an abundance of extracellular matrix (ECM). Diabetic wounds in mice treated with or without either the hUC-MSC tissue sheet, hUC-MSC injection, or fiber only revealed that hUC-MSC tissue sheet transplantation promoted diabetic wound healing with improved re-epithelialization, collagen deposition, blood vessel formation and maturation, and alleviated inflammation compared to that observed in other groups. Taken collectively, our findings suggest that hUC-MSCs cultured on PLGA scaffolds improve diabetic wound healing, collagen deposition, and angiogenesis, and provide a novel and effective method for cell transplantation, and a promising alternative for diabetic skin wound treatment.

## 1. Introduction

Diabetes mellitus is a serious and common chronic disease that poses a major threat to public health [1]. According to reports from the International Diabetes Federation (IDF), it is estimated that 537 million people had diabetes in 2021. This number will reach 643 million by 2030 and 783 million by 2045 [2]. Diabetic foot ulceration (DFU) is a common chronic diabetic complication [3,4]. Patients with diabetes have a 25% lifetime risk of developing foot ulcers compared to those without diabetes [5]. It is estimated that one patient with diabetes undergoes a lower limb amputation every 30 s worldwide [5]. DFU is a non-healing wound that is caused by an imbalance of various mechanisms, such as hemostasis, inflammation, collagen deposition, and angiogenesis [6]. In particular, the reduction of proangiogenic growth factors and decreased wound angiogenesis are the major reasons for impaired wound repair in diabetic patients [7]. Therefore, enhancing angiogenesis in the wounded area could be an effective strategy to accelerate wound healing.

In recent decades, growing evidence has demonstrated that mesenchymal stem cell (MSC)-based therapy can promote wound healing and inhibit scar formation [8,9,10]. Therefore, MSC transplantation is gaining increasing attention as a novel strategy for treating diabetic wounds [11]. Adipose-derived mesenchymal stromal cells (ADSCs) and bone marrow-derived mesenchymal stromal cells (BM-MSCs) exhibit similar effectiveness on wound healing—they can accelerate the diabetic wound healing via enhanced angiogenesis, promote cell proliferation, and ECM secretion [12,13]. However, several previous studies have indicated that the number and function of MSCs decrease with age or long-term disease. Because most diabetic patients are older adults, the majority of patients with diabetes lack sufficient and functional autologous stem cells for cell therapy of diabetic wounds [14,15,16,17]. Among the available sources of MSCs, the umbilical cord is economically viable and productive. Human umbilical cord-derived MSCs (hUC-MSCs) have been reported to accelerate wound healing; however, there are few studies on their effects on diabetes mellitus [18,19]. Moreover, the mechanism by which hUC-MSCs accelerate diabetic wound healing remains unclear.

The injection of a single-cell suspension into the periphery of the wounded area is a useful method for the delivery of MSCs [20]. However, cell suspension injection causes significant loss and uneven local distribution of cells, which reduces the effectiveness of the treatment [21]. Moreover, when the wound scale is large, peripheral injections cannot deliver the cells to the center of the wound. There are also some reports on intravenous injections; however, this method could reduce the delivery efficiency of MSCs to the wounded area because some transplanted cells will be entrapped in the lungs, leading to their accumulation [22,23].

Poly(lactic-co-glycolic acid) (PLGA) is a biodegradable and biocompatible copolymer material that has been used in a host of Food and Drug Administration (FDA) approved therapeutic devices [24]. In this study, we developed PLGA-based fibers as a scaffold to create hUC-MSC tissue sheets in vitro, which were transplanted in diabetic mice to assess their effects on diabetic wound healing. We optimized the conditions of scaffold preparation, cell culture, and tissue construction, and obtained 192 ± 14 μm thick tissue sheets. Before transplanting hUC-MSC tissue sheets on full-thickness excisional skin wounds of diabetic mice, the characteristics and properties of the tissue sheets were tested and verified. hUC-MSC tissue sheets effectively promoted epithelial regeneration, collagen deposition, infiltration of micro-vessels, and vascular maturation. These findings may help provide new clinically effective therapies for diabetic wounds.

## 2. Results

### 2.1. Characterization of hUC-MSCs

hUC-MSCs were maintained for 5–7 passages and their characteristics were confirmed based on the expression of MSC surface markers. Flow cytometry analysis demonstrated that hUC-MSCs were remarkably positive for CD73, CD90, CD105, and HLA-ABC, and remarkably negative for CD31, CD34, CD45, HLA-G, and HLA-DR (Figure 1a). The cells showed a typical spindle-like morphology under a bright-field microscope (Figure 1b). In addition, to examine whether hUC-MSCs retained their adipogenic, osteogenic, and chondrogenic differentiation potential, in vitro differentiation assays were conducted after the fifth passage. The cells were positive for Oil Red O staining (adipogenic marker) (Figure 1c), Alizarin S Red staining (osteogenic marker) (Figure 1d), and Alcian Blue staining (chondrogenic markers) (Figure 1e).

### 2.2. hUC-MSC Tissue Sheet Construction and Evaluation

The PLGA scaffold was used as a culture scaffold to guide tissue formation by hUC-MSCs (Figure 2a). To explore the importance of fiber orientation for the formation of hUC-MSC tissue sheets, we compared two types of PLGA scaffolds, a crossed structure and a traditional aligned structure. Compared with the crossed type, the thickness of the tissue sheet in the aligned type was more uniform and flatter (Appendix A). Therefore, we selected an aligned type of scaffold to create a tissue sheet (Figure 2b). The scanning electron microscope (SEM) image shows a parallel and evenly distributed construction (Figure 2c), and the diameter of each fiber was 3622 ± 563 nm (Figure 2d). Furthermore, to determine the optimal cell concentration for creating hUC-MSC tissue sheets, we cultured three different concentrations of MSCs (1 × 10^6^/2 × 10^6^ / 4 × 10^6^ per cm^2^) on the PLGA scaffold and then evaluated the thickness of the tissue sheet (Figure 2e,f). The results revealed that the thickness of the constructed tissue sheet increased with increasing seeded cell concentrations.

In addition, we performed a TUNEL immunofluorescence assay to detect the number of apoptotic hUC-MSCs in tissue sheets with different hUC-MSC concentrations. The percentage of TUNEL-positive MSCs increased to 26% in the tissue sheet group containing 4 × 10^6^ cells. In contrast, the 1 × 10^6^ and 2 × 10^6^ cell groups exhibited a high survival level (3.4 ± 2.2% and 5.7 ± 2.5%, 1 × 10^6^ cell group and 2 × 10^6^ cell group), and no significant difference was observed between the groups (Figure 2h,i). A higher proportion of collagen III aggregates in scarless fetal wound healing, whereas a higher proportion of collagen I is deposited in scarring adult wounds [25]. Therefore, to evaluate collagen deposition at different cell concentrations, the tissue sheets were immunostained for collagen I and III. The ratio of collagen I to collagen III in the 4 × 10^6^ cells group was higher than that in the 2 × 10^6^ cells group; however, there was no significant difference between the 1 × 10^6^ and the 2 × 10^6^ cell groups (Figure 2j,k). Therefore, we selected a cell concentration of 1 × 10^6^/cm^2^ to construct the tissue sheet and proceeded with the transplantation experiments.

### 2.3. hUC-MSC Tissue Sheet Accelerates Diabetic Wound Healing

We used the *db/db* diabetic mouse model to investigate the therapeutic potential of hUC-MSC tissue sheets in diabetic wound healing. Full-thickness skin wounds were created on the backs of the model mice, followed by treatment with an hUC-MSC tissue sheet, fiber-only, or hUC-MSC injection on the wound site. We first compared the hUC-MSC tissue sheet (1 × 10^6^ and 2 × 10^6^ cells) and control (left untreated) groups and observed that the tissue sheet-treated groups had remarkably accelerated wound healing. Interestingly, the average wound closure rate of the 1 × 10^6^ cells group was faster than that of the 2 × 10^6^ cell group, and a significant difference (*p* < 0.05) was observed on day 10 (Appendix A). Therefore, we used 1 × 10^6^ cells for subsequent transplantation experiments. Notably, the hUC-MSC tissue sheet group had a remarkably better wound closure effect than that observed in other groups on days 7, 10, and 14 after wounding. Moreover, the hUC-MSC injection group also had a better wound closure effect than the control group; however, only day 10 showed significant differences. Interestingly, the fiber-only group exhibited a slower healing rate than the control group, especially on day 7 (Figure 3a). In addition, H&E staining revealed that compared to other groups, the hUC-MSC tissue sheet group exhibited a thicker granulation tissue (Figure 3a,c), and had better wound re-epithelialization (Figure 3b,d).

### 2.4. hUC-MSC Tissue Sheet Induces Collagen Synthesis In Vivo

Regeneration of the dermis requires reconstruction of the collagen structure, mediated by fibroblasts and myofibroblasts [25]. Therefore, to further confirm the therapeutic effect of hUC-MSCs on diabetic wound healing, we investigated collagen deposition using Masson trichrome staining. As shown in Figure 4a,b, the wounded area in mice treated with the hUC-MSC tissue sheet or injections showed well-reorganized collagen fibers compared to the other groups; however, the hUC-MSC tissue sheet group exhibited a more complex collagen fiber structure with thicker collagen bundles than the injection group. IF staining (Figure 4c) of collagen I and III revealed that the wounded area treated with the hUC-MSC tissue sheet had a higher collagen III to collagen I ratio. This result was consistent with the data from the in vitro experiments (Figure 2j,k). Therefore, we speculated that using hUC-MSC tissue sheets would likely cause the diabetic wounds to progress to scarless healing.

### 2.5. hUC-MSC Tissue Sheet Remarkably Promotes New Blood Vessel Formation and Maturation in Diabetic Wounds

To further study the mechanism of wound healing promotion by hUC-MSC tissue sheets, we evaluated whether the therapeutic effect of hUC-MSC tissue sheets on wound healing is mediated by promoting angiogenesis. CD31, a specific marker for capillary endothelial cells, is widely used to evaluate injury-associated angiogenesis. A significantly (*p* < 0.01) increased number of CD31-positive cells was observed in the wound center of the hUC-MSC tissue sheet and injection groups compared to that in the control and fiber-only groups (Figure 5a,c). Furthermore, the number of CD31-positive cells in the hUC-MSC injection group was lower than that in the hUC-MSC tissue sheet group, suggesting that hUC-MSCs promote angiogenesis in the wound center, and hUC-MSCs cultured on the PLGA scaffold effectively enhanced this effect. Interestingly, there was no significant difference in the number of CD31-positive cells at the wound edge between the hUC-MSC tissue sheet and injection groups (Figure 5b,e). CD31-positive endothelial cells covered with α-SMA-positive mural cells are typically used to evaluate the matured vessels. Therefore, we further analyzed α-SMA expression in these groups. The density of matured vessels in the wound center was significantly increased in both the hUC-MSC tissue sheet and injection groups compared to other groups, whereas the density of matured vessels in the wound edge was markedly increased only in the hUC-MSC tissue sheet group both at the wound center and edge (Figure 5a,b,d,f). These results demonstrated that hUC-MSC tissue sheets could quickly and effectively promote the maturation of blood vessels in diabetic wound areas.

### 2.6. hUC-MSC Tissue Sheet Regulates the Inflammatory Response in Diabetic Wounds

To evaluate the effects of transplanted hUC-MSC tissue sheets on the infiltration of macrophages on day 14 after treatment, wound tissues were immunostained for CD68, which is a tissue macrophage marker. As shown in Figure 6a,b, a large number of CD68-positive cells were observed at the wound sites of diabetic mice. The number of CD68-positive cells in the wound site of the hUC-MSC tissue sheet and injection groups was significantly lower than that in the control group and fiber-only group. However, the large error bar in fiber-only group is because there were too few tissues available for quantitative statistics. Furthermore, no significant difference was observed between the two hUC-MSC groups. Collectively, these findings demonstrated that hUC-MSC tissue sheets or injections could decrease the number of macrophages in the local wound sites of diabetic mice, potentially contributing to the regulation of the inflammatory response and enhancing diabetic wound healing.

## 3. Discussion

Wound healing is a complex biological process that involves numerous molecules. Disruption of one or more stages during this process can lead to delayed or non-healing of the wound. In particular, some chronic or grave conditions, such as burns and diabetes, are associated with seriously impaired wound healing [26]. This impaired wound healing is mainly due to a reduction in growth factors, impaired angiogenesis, and reduced wound cell function [27]. Traditional treatment methods for skin injuries include adhesive wound dressings, hyperbaric oxygen therapy, negative pressure therapy, and autologous skin transplantation. However, these methods are limited [28]. Therefore, there is an urgent need to develop innovative and effective treatment strategies for patients with chronic and severe skin damage.

MSCs derived from different tissues can promote the healing of diabetic wounds [29,30]. UC-MSCs are derived from neonatal tissues and could be obtained using non-invasive methods. They have a greater proliferative capability, especially under hypoxic conditions [31]. Therefore, in our study, we selected hUC-MSCs for transplantation to accelerate diabetic wound healing.

The most commonly used method for cell transplantation is the injection of a single-cell suspension around the periphery of the wound area. However, owing to the lack of extracellular matrix (ECM) and intercellular communication between cells, this method often leads to a low survival rate of cells after transplantation and a limited therapeutic effect. In previous studies, our research group reported that in vitro construction of cells into tissues followed by transplantation greatly increased the cell survival rate in vivo and improved heart function and lower limb ischemia [24,32]. In this study, we used a fiber-based scaffold to construct an MSC tissue sheet with a 3D structure for transplantation. We hypothesized that tissue sheets could not only avoid the communication disruption between cells but also maintain the integrity of the ECM, while also increasing the number of transplanted cells through the 3D structure, resulting in better therapeutic effects. Our results confirmed this hypothesis and showed that compared with traditional single-cell injection, transplantation of hUC-MSC tissue sheets could accelerate the healing speed of diabetic ulcer wounds, induce angiogenesis and maturation of the center and edge in the wounded area, and improve their collagen remodeling (Figure 4). Interestingly, we found that transplantation of fibers alone affected the healing rate of diabetic wounds but had no significant adverse effects on re-epithelialization, angiogenesis, or collagen remodeling (Figure 3, Figure 4 and Figure 5). This is possibly because, without cells, the fiber could be directly in contact with and adhere to the affected part, preventing the contraction of the skin during the wound-healing process. Reducing the fiber density may solve this problem. Moreover, as diabetic wounds in humans do not shrink like those in mice, this problem may be negligible in the process of human diabetic wound healing.

In addition, we evaluated the effect of the number of transplanted cells in the tissues and found that the tissue sheets constructed with 4 × 10^6^ cells had a higher rate of apoptosis than that of the 1 × 10^6^ and 2 × 10^6^ cell groups (Figure 2). This is because the thickness of the tissue exceeds the diffusion limits of nutrients and oxygen [33]. Although there was no significant difference in the apoptotic rate between tissues with 1 × 10^6^ cells and 2 × 10^6^ cells in vitro, the group transplanted with 1 × 10^6^ cells showed a more pronounced therapeutic effect than the 2 × 10^6^ cell group in vivo (Appendix A). This may be due to the harsh environment of the wounded area, especially in the early stage, which is accompanied by a large amount of tissue fluid exudation. The tissue fluid also contains a large number of inflammatory factors, which could affect the vascular construction between the graft and the host. If the tissue exceeds the appropriate thickness, the nutrient supply of the graft might be affected, thereby affecting the therapeutic effect in wound healing and potentially leading to a more severe inflammatory response. Therefore, it is important to carefully select the number of transplanted cells and the thickness of the tissues to induce the desired therapeutic effect.

The typical process of skin regeneration is divided into four overlapping phases: inflammation, angiogenesis, cell proliferation, and wound remodeling. During the cell proliferation phase, angiogenesis, collagen deposition, re-epithelialization, and wound contraction occur simultaneously. Angiogenesis is crucial for wound healing and tissue repair [34]. Our data demonstrated that hUC-MSC tissue sheets promote angiogenesis in both the wound area and edge. Furthermore, the hUC-MSC tissue sheet also had an obvious effect on vascular maturation compared to single-cell injection (Figure 5). The hUC-MSC tissue sheets not only accelerated wound healing but also effectively promoted re-epithelialization inside the wound area (Figure 3). Therefore, our transplantation strategy not only allows cells to directly contact the ulcer wound but also allows the secreted factors of cells to act directly on the wound. In addition, this alternative strategy to single-cell transplantation could protect the integrity of the ECM between cells, thereby reducing the cell loss rate, improving cell survival after transplantation, and producing more factors that contribute to angiogenesis. Moreover, the remodeling phase of a wound is between 2 weeks to more than 1 year, and is closely related to the production and reorganization of the ECM; therefore, it has an important impact on the extent of scarring [35]. In our study, both the in vitro and in vivo results showed remarkable consistency. A large amount of ECM deposition was observed in the tissue sheet and transplantation site, and the proportion of collagen III was significantly higher than that of collagen I (Figure 2 and Figure 4). These results indicate that the transplanted tissue sheets could inhibit scarring in the wound area.

Chronic tissue inflammation is a recognized feature of diabetes mellitus [36]. Therefore, improvement of the microenvironment of the wound area is a key factor in wound healing and tissue remodeling [37]. Additionally, MSCs can play anti-inflammatory and immunoregulatory roles [38]. Therefore, we investigated the immunoregulatory function of hUC-MSC tissue sheets in the microenvironment of diabetic wounds and found that hUC-MSC tissue sheets regulate the wound microenvironment by reducing the infiltration of macrophages. However, there was no significant difference between the hUC-MSC tissue sheet and injection groups (Figure 6). This phenomenon may be because the two groups had the same number of cells. Nevertheless, tissue repair and reconstruction require a certain degree of inflammatory responses to recruit more anti-inflammatory, proangiogenic, and other factors to the wound area to promote wound healing.

Furthermore, the long-term outcomes and the effectiveness of MSC-based cell therapies still need to be improved. Maksimova’s and Zhang’s studies showed that topical administration of MSCs is a safe and promising option, but it took effect mainly in the initial wound closure at an early stage [39,40]. Here, we provide a new method to enhance the effectiveness of MSCs and maybe improve the long-term outcomes in the future.

## 4. Materials and Methods

### 4.1. Construction of PLGA Scaffold

For the construction of the scaffold, PLGA (75/25; Sigma-Aldrich, St. Louis, MO, USA) was mixed with hexafluoro-2-propanol (HFIP, Wako Pure Chemical Industries, Tokyo, Japan) in a centrifugal tube (1.2 g:3 mL, *w*/*v*), and an automated electrospinning machine (NF-103, MECC, Fukuoka, Japan) was used to synthesize the fibers. The mixed solution was loaded into a 3 mL syringe to which a needle with a 0.6 mm inner diameter was attached to connect to the positive electrode of the high-voltage power supply (10 kV). A layer of aluminum foil was attached to the grounded drum. The drum was rotated at a speed of 1000 rpm and used to collect the PLGA scaffold. The distance between the needle tip and the drum was maintained at 15 cm. The spinning process lasted for 120 min (aligned type: same direction for a total of 120 min; crossed type: 60 min for parallel direction (0 degrees) and 60 min for vertical direction (90 degrees)). The fiber sheet was then transferred to a polydimethylsiloxane (PDMS) frame (1 cm × 1 cm) for subsequent cell seeding. After construction, the fibers were examined using a scanning electron microscope, as previously described [24].

### 4.2. Culture of hUC-MSCs

hUC-MSCs were provided by Cell Exosomes Therapeutics Co., Ltd. (Tokyo, Japan), and all experiments were approved by the Ethical Committee of Osaka University. Briefly, the cells were cultured in MSCs Xeno-Free culture medium (Takara Bio Inc., Shiga, Japan) at 37 °C in a 5% CO_2_ incubator. The culture medium was changed every 2–3 d. Upon reaching 80–90% confluence, the cells were detached using TrypLE Select (Gibco, Waltham, MA, USA) for further expansion. All experiments used cells in passages 5–7.

### 4.3. Characterization of hUC-MSCs

Immunophenotyping was performed using flow cytometry (FACS II). Briefly, after seven passages, hUC-MSCs were harvested and dissociated into single cells, washed with phosphate-buffered saline (PBS), and stained with the following fluorescence-conjugated antibodies (Table 1): anti-hCD31-PE, anti-hCD34-PE, anti-hCD45-PE, anti-hCD73-PE, anti-hCD90 (Thy1)-PE, anti-hCD105-PE, anti-HLA-G-PE, anti-HLA-DR-PE, and anti-HLA-ABC-PE. Mouse IgG1 κ isotype was used to stain the cells as a control. The antibodies were purchased from BioLegend (San Diego, CA, USA). Data were analyzed using FlowJo software v 10.5.3 (BD, Franklin Lakes, NJ, USA).

The adipogenic, osteogenic, and chondrogenic differentiation potentials of hUC-MSCs were evaluated using a differentiation medium (PromoCell, Heidelberg, Germany) for 2, 2, and 3 weeks, respectively. Oil Red O, Alizarin Red S, and Alcian Blue (Sigma-Aldrich) staining were performed to confirm the differentiation potential of hUC-MSCs.

### 4.4. hUC-MSC Tissue Sheet Formation

hUC-MSCs were seeded onto a PLGA scaffold (1–4 × 10^6^ cells/cm^2^). iMatrix-511 (Matrixome, Osaka, Japan) was added during cell seeding at a concentration of 10 μg/mL. The samples were then cultured in a 5% CO_2_ humidified atmosphere at 37 ℃ for 3–5 d before transplantation, and the medium was changed every 2 days.

### 4.5. In Vivo Wound Healing Experiments in a db/db Mouse Model

Animal experiments were performed according to the guidelines of Osaka University, and all experiments using hUC-MSCs were approved by the Ethical Committee of Osaka University (20262(T2)-2). All experiments were performed using male *db/db* (*BKS.Cg-+Leprdb/+Leprdb/Jcl*) mice (10 weeks old, 46.2 ± 1.8 g weight; CLEA Japan, Inc., Shizuoka, Japan). Blood glucose levels were monitored and the mice (*n* = 28) showing blood glucose levels greater than 300 mg/dL were diagnosed as diabetic. The diabetic mice were kept under observation for 10 days before skin wound creation.

Animals were anesthetized using isoflurane (1.5%; Mylan Inc., Canonsburg, PA, USA). After shaving the dorsal hair of the mice, two full-thickness excisional skin wounds (8 mm in diameter) were created on their backs. The mice were then randomly divided into four different groups (seven mice per group): (1) left untreated (control); (2) treated with 1 × 10^6^ hUC-MSCs cultured on PLGA scaffold (hUC-MSC tissue sheet); (3) PLGA scaffold-treated (Fiber only); and (4) 1 × 10^6^ hUC-MSC suspension in 50 μL PBS subcutaneous administration (hUC-MSC injection). In the injection group, mice were subcutaneously injected with hUC-MSCs in PBS at five injection sites (10 μL per site). In hUC-MSC tissue sheet group and Fiber only group, the PDMS frame of the PLGA scaffold was cut and removed, and the PLGA sheet was placed on the wound area. After treatment, a transparent and semi-occlusive adhesive dressing (Tegaderm; 3M, Saint Paul, MO, USA) was applied to protect the wounds. The mice were housed individually. The wound area was photographed and measured on days 0, 3, 7, 10, and 14 after wounding. The pictures were then analyzed using image ImageJ software (National Institutes of Health, Bethesda, MD, USA). Mice were sacrificed on day 14 after surgery, and wound skin tissues were harvested for histological analysis.

### 4.6. Immunofluorescence and Histological Analysis

hUC-MSC tissue sheets were fixed in 4% paraformaldehyde overnight at 4 °C to create frozen sections. TUNEL assay was performed using the Click-IT TUNEL kit (Alexa Fluor 647), following the manufacturer’s instructions (Invitrogen; Thermo Fisher Scientific, Waltham, MA, USA). Immunohistology was performed. The sections were incubated with primary antibodies (Table 1) overnight at 4 °C, washed with PBS, and then incubated with the respective secondary antibodies at 37 °C for 1 h. After counterstaining with 2-(4-amidinophenyl)-1H-indole-6-carboxamidine (DAPI) (Invitrogen) or Hoechst 33342 (Invitrogen), the sections were analyzed using a fluorescence microscope (BZ-X800, KEYENCE, Osaka, Japan) and a Nikon A1 confocal microscope (Nikon, New York, NY, USA)

Skin samples from the wound site of each group of mice were fixed in formalin, transferred to ethanol, and embedded in paraffin. Serial sections were prepared at a 5-μm thickness from the middle of the wound. The sections were stained with hematoxylin and eosin (H&E) and Masson’s trichrome. For histological analysis, wound sections were stained with H&E and the percentage of re-epithelialization (rE%) was assessed as
rE% = (Wt/Wo × 100)%,(1)
where Wo means the original wound length and Wt represents the length of newly generated epithelium across the surface of the wound.

Masson’s trichrome staining was used to evaluate the degree of collagen maturity. For immunohistology of wound tissue, dewaxed paraffin sections were washed in PBS, and antigen retrieval was performed in Target Retrieval Solution (pH = 6; DAKO Japan, Inc., Tokyo, Japan) at 121 °C for 10 min. Sections were stained as described above. Furthermore, to quantify the number of mature blood vessels, CD31, α-SMA, and cell nuclei were stained orange, green, and blue, respectively. Orange and green double staining represent mature blood vessels, as reported previously [41,42,43].

### 4.7. Statistical Analysis

All quantitative data are presented as mean ± standard deviation (SD). Analysis of variance (ANOVA) or *t*-test was used to evaluate statistical significance, which was defined by a *p*-value < 0.05 (significance was set at * *p* < 0.05, ** *p* < 0.01, *** *p* < 0.001).

## 5. Conclusions

In summary, our study demonstrated that hUC-MSC tissue sheets could effectively accelerate the rate of wound healing, improve re-epithelialization of wound sites, promote collagen deposition and remodeling, enhance angiogenesis and vessel maturation, and regulate the immune microenvironment of the wounded area. Our findings provide a new and effective method for cell transplantation and a new strategy for the treatment of not only diabetic skin wounds but also burns and traumatic skin injuries.

## Figures and Tables

**Figure 1 ijms-23-12697-f001:**
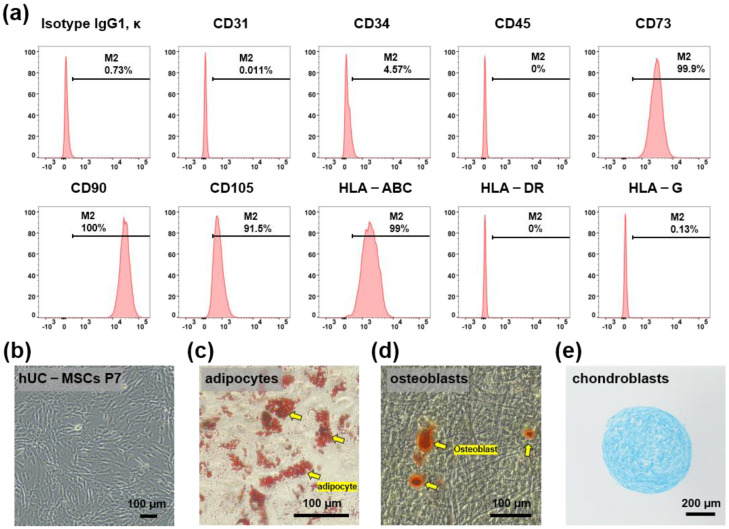
Characterization of human umbilical cord mesenchymal stem cells (hUC-MSCs). (**a**) Flow cytometry analysis using hUC-MSCs surface markers CD31, CD34, CD45, CD73, CD90, CD105, HLA-ABC, HLA-DR, and HLA-G. (**b**) Bright-field microscopic images show spindle-like hUC-MSCs. Scale bar = 100 μm. (**c**–**e**) Differentiation ability of hUC-MSCs. Adipocytes, osteoblasts, and chondrocytes were detected using Oil Red O, Alizarin Red, and Alcian Blue, respectively, scale bar = 100 μm, 100 μm, and 200 μm, respectively.

**Figure 2 ijms-23-12697-f002:**
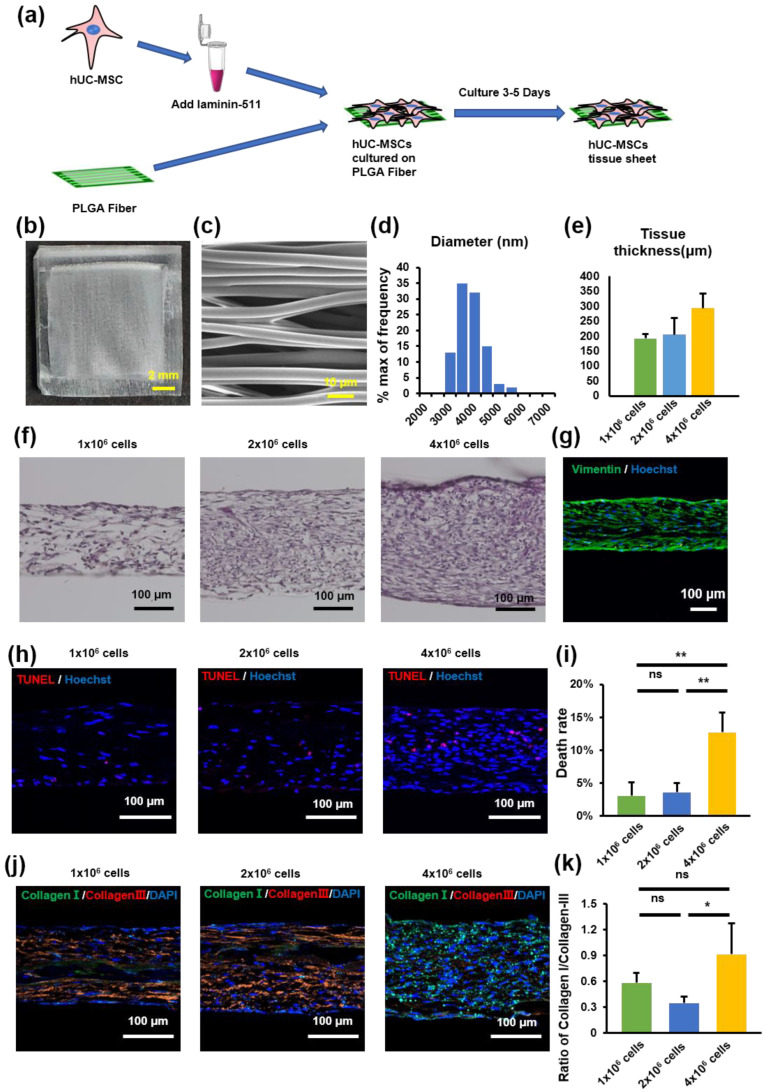
Human umbilical cord mesenchymal stem cell (hUC-MSC) tissue sheet construction and evaluation. (**a**) Schematic representation of tissue construction in vitro. (**b**) Image of PLGA-aligned fibers. Scale bar = 2 mm. (**c**) Scanning electron microscopy image of PLGA-aligned fibers. Scale bar = 10 μm. (**d**) Quantitative analysis of the diameter of fibers in PLGA-aligned fibers. (**e**) Quantitative analysis of the thickness of hUC-MSC tissue sheets with different cell densities in (**f**). (**f**) Hematoxylin and eosin staining of hUC-MSC tissue sheets with different cell densities. Scale bar = 100 μm. (**g**) Immunohistological images of Vimentin (green) in hUC-MSC tissue sheet (1 × 10^6^ cells). Scale bar = 100 μm. (**h**) TUNEL staining of hUC-MSC tissue sheet sections with different cell densities. Scale bar = 100 μm. (**i**) Quantitative analysis of cell death rate in different groups. (**j**) Immunohistological images of collagen I (green) and collagen III (red) in hUC-MSC tissue sheet section with different cell densities. Scale bar = 100 μm. (**k**) Quantitative analysis of the ratio of Collagen I/Collagen III in different groups. Results are presented as mean ± SD. Significance was determined using analysis of variance. ANOVA. * *p* < 0.05, ** *p* < 0.01, ns means no statistical significance. PLGA, poly(lactide-co-glycolide).

**Figure 3 ijms-23-12697-f003:**
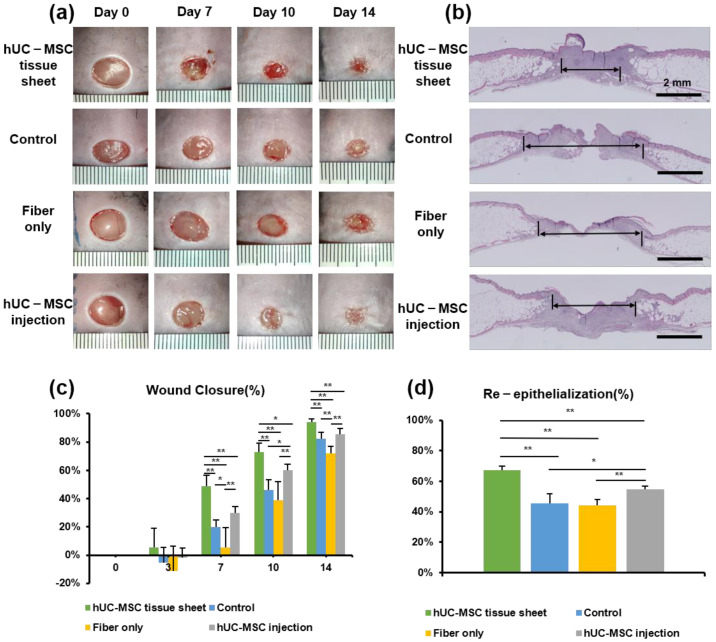
Human umbilical cord mesenchymal stem cell (hUC-MSC) tissue sheet accelerates diabetic wound healing. (**a**) Representative photographs of full-thickness excision wounds at 0, 7, 10, and 14 days after wounding. (**b**) Hematoxylin and eosin staining of wound sections treated with hUC-MSC tissue sheet, fiber only, and hUC-MSCs injection 14 days after wounding. The double-headed black arrows indicate the edge of the scars. Scale bar = 2 mm. (**c**) Quantitative analysis of the rate of wound closure in each group. (**d**) Quantitative analysis of the extent of re-epithelialization in (**b**). Results are presented as mean ± SD. Significance was determined using analysis of variance ANOVA. * *p* < 0.05, ** *p* < 0.01.

**Figure 4 ijms-23-12697-f004:**
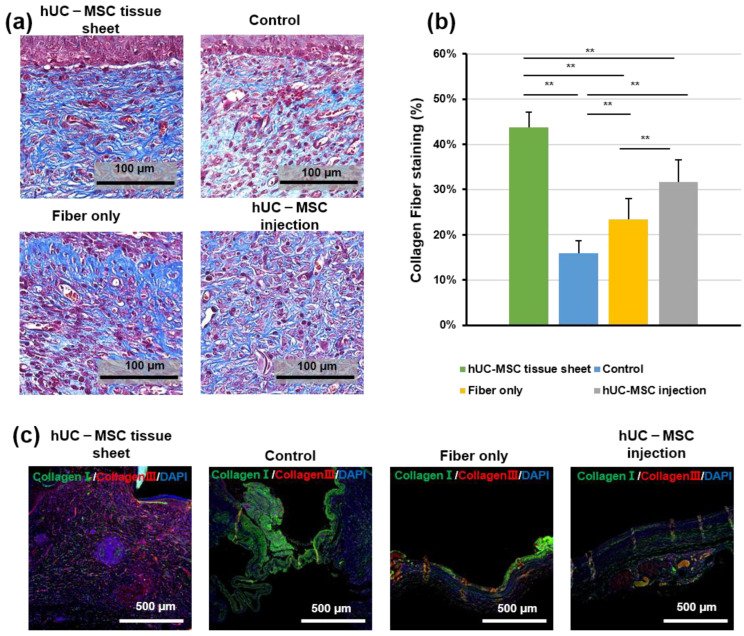
Human umbilical cord mesenchymal stem cell (hUC-MSC) tissue sheet induces collagen synthesis in vivo. (**a**) Masson staining of wound sections in different groups on day 14 after wounding. Scale bar = 100 μm. (**b**) Quantitative analysis of collagen fiber deposition in different groups 14 days after wounding. (**c**) Immunohistological analysis of collagen I (green) and III (red). Scale bar = 500 μm. Results are presented as mean ± SD. Significance was determined using analysis of variance ANOVA. ** *p* < 0.01.

**Figure 5 ijms-23-12697-f005:**
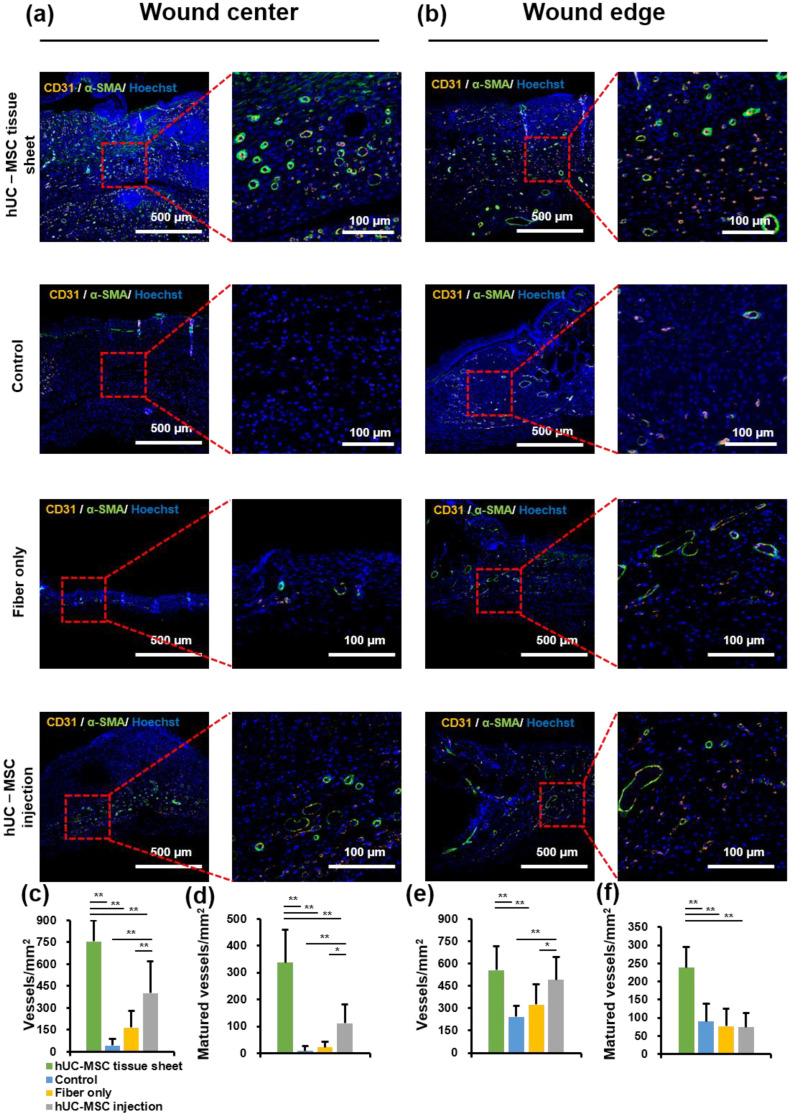
Human umbilical cord mesenchymal stem cell (hUC-MSC) tissue sheet remarkably promotes new blood vessel formation and maturation in diabetic wounds. (**a**) Representative images of tissue sections from the center of the diabetic wound area 14 days after wounding (scale bar = 500 μm and 100 μm). (**b**) Representative images of tissue sections from the edge of the diabetic wound area on day 14 after wounding (scale bar = 500 μm and 100 μm). (**c**) Quantitative analysis of vessel density of wound center in different groups 14 days after wounding. (**d**) Quantitative analysis of matured vessel density of wound center in different groups 14 days after wounding. (**e**) Quantitative analysis of vessel density of wound edge in different groups 14 days after wounding. (**f**) Quantitative analysis of matured vessel density of wound edge in different groups 14 days after wounding. Results are presented as mean ± SD. Significance was determined using analysis of variance ANOVA. * *p* < 0.05, ** *p* < 0.01.

**Figure 6 ijms-23-12697-f006:**
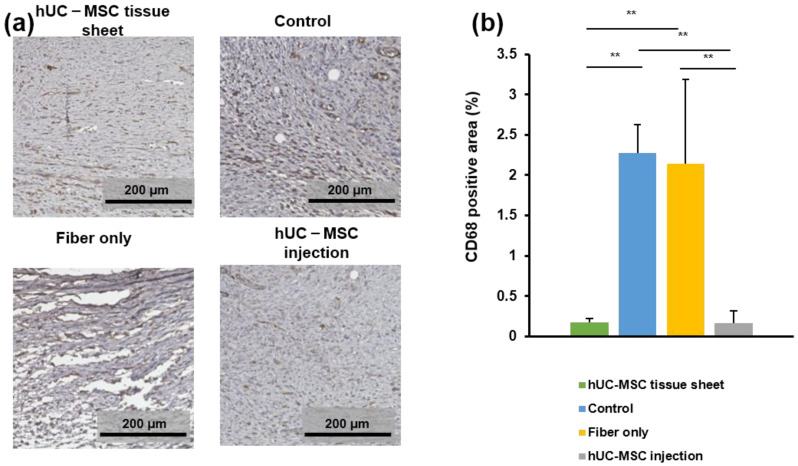
The human umbilical cord mesenchymal stem cell (hUC-MSC) tissue sheet regulates inflammation response in diabetic wounds. (**a**) Representative images of immunohistochemical sections in different groups on day 14 after wounding. Scale bar = 200 μm. (**b**) Quantitative analysis of immunohistochemical signals of CD68 in high power field images. In the fiber-only group, only the non-fiber area is analyzed. Results are presented as mean ± SD. Significance was determined using analysis of variance ANOVA. ** *p* < 0.01.

**Table 1 ijms-23-12697-t001:** The list of antibodies used in this study.

Antibody Name	Dilution	Company	Catalog #
hCD31	1:1000	Biolegend	303106
hCD34	1:1000	Biolegend	343506
hCD45	1:1000	Biolegend	304008
hCD73	1:1000	Biolegend	344004
hCD90	1:1000	Biolegend	328110
hCD105	1:1000	Biolegend	323206
HLA-ABC	1:1000	Biolegend	311406
HLA-DR	1:1000	Biolegend	307606
HLA-G	1:1000	Biolegend	335905
Vimentin [D21H3]	1:100	Cell Signaling	5741S
Collagen Type Ⅰ	1:100	Sigma-Aldrich	C2456
Collagen Type Ⅲ	1:100	Abcam	ab7778
CD31	1:100	Abcam	ab28364
Actin (Smooth Muscle)	1:100	Dako	M0851
CD68	1:100	Abcam	ab31630

## Data Availability

All data generated and/or analyzed in this study are included in this published article.

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
