# Peer review of "Tissue Sheet Engineered Using Human Umbilical Cord-Derived Mesenchymal Stem Cells Improves Diabetic Wound Healing"

_ijms, 2022, doi:10.3390/ijms232012697_

Round 1
Reviewer 1 Report
The presented paper discovers that tissue sheets engineered using human umbilical cord-derived mesenchymal stem cells improve diabetic wound healing in mouse models. The authors optimized the conditions of scaffold preparation, cell culture, and tissue construction, and obtained ~200 μm thick tissue sheets.
1. Authors need to define the mature vessels in the Materials and Methods. How do mature vessels differ from innate vessels?
2. Figure 3: Figure 3c presented the quantitative analysis of the wound closure in each group as the QUADRATIC DEGREE of the size of the ulcer, and the Figure 3d presented the quantitative analysis of the extent of re-epithelialization as the LINEAR DEGREE of the size of the ulcer. The graph's dimensions should be unified.
3. The authors should uncover the effectiveness of mesenchymal stromal cells from various sources for the healing in the Introduction section:
3.1. Krasilnikova, O.A.; Baranovskii, D.S.; Lyundup, A.V.; Shegay, P.V.; Kaprin, A.D., Klabukov, I.D. Stem and Somatic Cell Monotherapy for the Treatment of Diabetic Foot Ulcers: Review of Clinical Studies and Mechanisms of Action. Stem Cell Reviews and Reports 2022, 18, 1974–1985. https://doi.org/10.1007/s12015-022-10379-z
3.2. Du, S., Zeugolis, D. I., & O’Brien, T. (2022). Scaffold-based delivery of mesenchymal stromal cells to diabetic wounds. Stem Cell Research & Therapy, 13(1), 1-19. https://doi.org/10.1016/j.jcyt.2021.08.001
4. The long-term outcomes and issue of enhancing the effectiveness of MSC-based cell therapies should be disclosed in the Discussion section:
4.1. Zhang, C.; Huang, L.; Wang, X.; Zhou, X.; Zhang, X.; Li, L.; ... & Zhou, X. Topical and intravenous administration of human umbilical cord mesenchymal stem cells in patients with diabetic foot ulcer and peripheral arterial disease: a phase I pilot study with a 3-year follow-up. Stem cell research & therapy 2022, 13(1), 1-14. https://doi.org/10.1186/s13287-022-03143-0
4.2. Maksimova, N.V.; Michenko, A.V.; Krasilnikova, O.A.; Klabukov, I.D.; Gadaev, I.Y.; Krasheninnikov, M.E.; Belkov, P.A.; Lyundup, A.V. Mesenchymal stromal cell therapy alone does not lead to complete restoration of skin parameters in diabetic foot patients within a 3-year follow-up period. BioImpacts 2022, 12(1), 51-55. https://doi.org/10.34172%2Fbi.2021.22167
Reviewer 2 Report
The study is showing the regenerative application of hUC-MSCs in diabetes wound treatment using rodent model. The manuscript is well written and designed. Some inaccuracies were observed.
Remarks.
1. The model by using hUC-MSCs and rodent animals is causing some doubts in terms of biocompatibility, but is a model system to investigate human cells in vivo.
2. Line 77. The term „direct differentiation assay” is not clear. In that case what is indirect differentiation?
3. Line 91. Description of scaffold structure „vertical structure and a traditional parallel structure“ should be corrected. The structures are usually „linear“ or “random“, or “aligned”, or other types, but how to understand „vertical“ ? Was this scaffold always used in vertical position? But Figure 2A are showing a horizontal position of used scaffold for cell seeding.
4. The Fig.2 j. The last micrograph is missing color identification. J and k are without brackets.
5. Lines 104-105. The sentence needs corrections: In contrast, the 1×106 and 2×106 cell groups exhibited a high survival level (3.4 104± 2.2% vs 5.7 ± 2.5% vs 26.1 ± 16.8 %), and no significant difference was observed between the groups (Figure 2h, i).” The authors mentioned two groups, but in the brackets there are three groups data that differ significantly.
6. Fig. 3. If the titles of pictures are in the middle, the two titles (control and fiber only) in (a) does not give additional information just make confusion and should be eliminated.
7. In some legends there is written “immunostaining”, it should be “immunohistochemical” since it is in tissues.
8. The method part “Construction of PLGA scaffold“ is missing description of generation of different PLGA structures („vertical“ and „parallel“, which should be changed to more understandable). In scaffold description “aligned” is more acceptable than “parallel”. The “vertical” is not clear at all.
9. The method part is missing immunohistological part.
10. Nothing was mentioned about the bioethical permission to works with animals.
